# Epidemiological Characteristics of Deaths from COVID-19 in Peru during the Initial Pandemic Response

**DOI:** 10.3390/healthcare10122404

**Published:** 2022-11-30

**Authors:** Willy Ramos, Juan Arrasco, Jhony A. De La Cruz-Vargas, Luis Ordóñez, María Vargas, Yovanna Seclén-Ubillús, Miguel Luna, Nadia Guerrero, José Medina, Isabel Sandoval, Maria Edith Solis-Castro, Manuel Loayza

**Affiliations:** 1Centro Nacional de Epidemiología, Prevención y Control de Enfermedades, Ministerio de Salud, Lima 15072, Peru; 2Instituto de Investigaciones en Ciencias Biomédicas (INICIB), Universidad Ricardo Palma, Lima 15039, Peru; 3Programa de Especialización en Epidemiología de Campo (PREEC), Lima 15072, Peru; 4Unidad de Post Grado, Facultad de Medicina de San Fernando, Universidad Nacional Mayor de San Marcos, Lima 15001, Peru; 5Departamento Académico de Medicina Humana, Facultad de Ciencias de la Salud, Universidad Nacional de Tumbes, Tumbes 24001, Peru

**Keywords:** coronavirus, COVID-19, deaths, epidemiology, Peru

## Abstract

Background and aim: Peru is the country with the highest mortality rate from COVID-19 globally, so the analysis of the characteristics of deaths is of national and international interest. The aim was to determine the epidemiological characteristics of deaths from COVID-19 in Peru from 28 March to 21 May 2020. Methods: Deaths from various sources were investigated, including the COVID-19 Epidemiological Surveillance and the National System of Deaths (SINADEF). In all, 3851 deaths that met the definition of a confirmed case and had a positive result of RT-PCR or rapid test IgM/IgG, were considered for the analysis. We obtained the epidemiological variables and carried out an analysis of time defined as the pre-hospital time from the onset of symptoms to hospitalization, and hospital time from the date of hospitalization to death. Results: Deaths were more frequent in males (72.0%), seniors (68.8%) and residents of the region of Lima (42.7%). In 17.8% of cases, the death occurred out-of-hospital, and 31.4% had some comorbidity. The median of pre-hospital time was 7 days (IQR: 4.0–9.0) and for the hospital time was 5 days (IQR: 3.0–9.0). The multivariable analysis with Poisson regression with robust variance found that the age group, comorbidity diagnosis and the region of origin significantly influenced pre-hospital time; while sex, comorbidity diagnosis, healthcare provider and the region of origin significantly influenced hospital time. Conclusion: Deaths occurred mainly in males, seniors and on the coast, with considerable out-of-hospital deaths. Pre-hospital time was affected by age group, the diagnosis of comorbidities and the region of origin; while, hospital time was influenced by gender, the diagnosis of comorbidities, healthcare provider and the region of origin.

## 1. Introduction

The current COVID-19 pandemic, reported since December 2019, constitutes the biggest public health problem in decades. This scenario includes the diversity of economic, social, and demographic aspects and different health system responses to identify and provide medical care to affected people [1,2,3]. The evolution of the pandemic, in terms of incidence, mortality and the speed of expansion, is heterogeneous, with differences between countries and even between regions of the same country [1,4,5].

Since the introduction of the pandemic in Peru, epidemiological surveillance was implemented, including the notification of deaths, a useful tool to measure the impact of COVID-19. Mortality is an important indicator in pandemic circumstances. The diagnostic testing strategy can influence the number of deaths from COVID-19, health system responsiveness, the social behavior of the population, and the dynamic situation between countries, among other factors [3,6,7,8].

The Republic of Peru has 32,495,500 inhabitants, distributed in 24 regions, and one constitutional province. It is grouped into three natural regions: the coast located between the Peruvian sea and the Andes mountain range; the Andean region located in the Andes mountain range; and the jungle, which encompasses the Peruvian Amazon [9]. Lima is the capital of Peru, and it is located on the coast, with a population that exceeds 10 million inhabitants [10].

According to the Global Aging Index [11], created in 2015 by HelpAge International, Peru is ranked number 48 out of 96 and has a life expectancy of 76.5 years. The Encuestas Demográficas y de Salud Familiar (ENDES) and health analyses show that the Peruvian population has a health profile dominated by noncommunicable diseases, such as high blood pressure, cardiovascular diseases, cancer, diabetes mellitus and comorbidities that constitute risk factors for COVID-19. People aged 80 years or older have the highest comorbidity rates (67.6%) [11,12,13,14].

In 20 May 2020, the Centro Nacional de Epidemiología, Prevención y Control de Enfermedades del Ministerio de Salud del Perú had reported 108,769 confirmed cases of COVID-19, 3148 deaths, and with a case fatality rate of 2.89%. At that time, there was a high demand for people who required health services and insufficient supply, which caused the health establishments to be overwhelmed by a large number of cases. At the same time, the Ministry of Health tried to organize the response with available resources, which included: mandatory social immobilization (quarantine); the decentralization of molecular test processing; the implementation of temporary healthcare facilities for COVID-19 cases; an increase in hospital beds and intensive care unit beds (ICU beds); acquiring and implementing oxygen plants; acquiring mechanical ventilators; as well as management for the future acquisition of vaccines against COVID-19 [15,16].

By the end of August 2020, Peru became the country with the highest mortality rate from COVID-19 globally, reaching 89.4 deaths per one hundred thousand inhabitants, so the analysis of the characteristics of deaths is of national and international interest [17]. The objective of this research was to determine the epidemiological characteristics of deaths from COVID-19 in Peru, from 28 March to 21 May 2020, 85 days after the report of the first confirmed case.

## 2. Methods

This was a case series study. The epidemiological surveillance database of coronavirus disease was revised, including the deaths investigated by the Epidemiology Offices personnel of the notifying health facilities at the national level. The quality control and monitoring of the notification were carried out by epidemiology personnel at the national level, sector workers assigned to each region, and Regional Health Directorates.

Confirmed deaths from COVID-19 were considered for analysis and included only those with a positive (reactive) RT-PCR result, rapid tests, or both, that were notified between 28 March and 21 May 2020. Deaths not attributable to COVID-19 were excluded from the study; that is, deaths caused by entities or processes in which there was no evidence that COVID-19 infection changed the course of the disease. The sources of information on deaths from COVID-19 were as follows:-National System of Deaths (SINADEF);-Database of RT-PCR test results (Netlab) and/or rapid tests (SISCOVID);-Clinical histories of hospitalized cases;-If complete information on death was not available from these sources, telephone communication was made with the epidemiologists of the regions, who performed a verbal autopsy in some cases.

From these sources, epidemiological variables such as age, gender, the region of residence (probable place of infection), date of the onset of symptoms, the date of hospitalization, the date of death, place where the death occurred, health provider institution, and the diagnosis of comorbidities were extracted. The following age groups were considered: children and adolescents (0–17 years); youth (18–29 years); adults (30–59 years); and seniors (60 years or older). In comorbidities, diabetes mellitus, arterial hypertension, cardiovascular diseases, cancer, chronic kidney disease, obesity, chronic pulmonary and bronchial diseases, were considered. The mortality rate in Peru from COVID-19 was obtained according to the four regions of origin of the patient: Lima, the rest of the coast, the Andes mountain range, and the jungle.

A time analysis was performed. The pre-hospital time was defined as the time from the onset of symptoms to the date of hospitalization, and the time in hospital was defined as the time from hospitalization to death. Finally, it was analyzed whether the pre-hospital and hospital times showed differences according to the epidemiological characteristics evaluated. The analysis was guided under the assumption that a longer pre-hospital time indicates a delay in reaching a health facility, and that a longer hospital time is an indicator of a higher survival of the cases.

The statistical analysis was carried out with the statistical program SPSS version 25 for Windows. Descriptive statistics were performed based on obtaining frequencies, percentages, measures of central tendency, and dispersion. Bivariate statistics were performed with Pearson’s chi-square test for comparison of proportions as well as student’s *t*-tests and one-way ANOVA for comparison of means. The bivariate analysis of the variables pre-hospital time and hospital time was performed with non-parametric tests such as the Kruskal-Wallis test and the Mann-Whitney U test, in the case of the Kruskal-Wallis test, a posteriori analysis was performed with the Dunn-Bonferroni test. The multivariable analysis to model pre-hospital and hospital time was carried out with Poisson regression with robust variance [18,19]. The calculations were performed with a confidence level of 95%.

The protocol was approved by the Ethics Committee of the Faculty of Medicine of the Ricardo Palma University, Lima-Peru (Expedited review, Code PI-021-2020). Confidentiality of the information collected was guaranteed and was used only for the study. Informed consent was not required as the research was based on secondary sources.

## 3. Results

During the study period, 3920 COVID-19 confirmed deaths were registered. In all, 11 of those deaths were excluded because of incidental causes (acute myocardial infarction, ruptured cerebral aneurysm, diabetic ketoacidosis, tuberculous meningoencephalitis, sepsis of dermal focus, and perforated appendicitis), and 58 due to incomplete data, leaving 3851 deaths available for analysis. Of the 3851 deaths, 96.4% corresponded to pneumonia, while the remaining 3.6% corresponded to home deaths of cases with very fast-evolving respiratory symptoms not explained by another disease.

### 3.1. Epidemiological Characteristics

As of 21 May 2020, the number of COVID-19 confirmed deaths in Peru showed an upward trend; however, there was a drop in the number of deaths in the last week of the study, due to the duration of the investigation delay of confirmatory results of RT-PCR tests, which were later regularized.

The highest frequency of confirmed deaths from COVID-19 corresponded to males (72.0%), adults over 60 years of age or older (68.8%), and people from the region of Lima (42.7%), observing that 31.4% had some comorbidity. In 79.8% of cases, the death occurred in a public hospital; while, 17.8% died at home, in accommodation, shelter, on a public road, or in transit to a hospital. This is shown in Table 1.

The mortality rate from COVID-19, between 28 March and 21 May 2020, was 11.8 per 100,000 inhabitants, this being higher in the areas of the rest of the coast (18.1 per 100,000 people) and in Lima (14.0 per 100,000 people). This is shown in Table 2.

### 3.2. Stratified Analysis

The stratified analysis showed that the average age of women was significantly higher than that of men (Student *t*-test; *p* = 0.002). A significant difference was also observed in the proportion of comorbidities according to the age group (Pearson’s chi square Test; *p* = 0.020); thus, the deceased between 0–17 years of age had a lower proportion of comorbidities and these increased with age to reach 36.9% in people 60 years or older. The mean age showed no statistically significant difference according to the region of origin (One-way ANOVA; *p* = 0.141) (Table 3).

### 3.3. Analysis of Pre-Hospital and Hospital Times

#### 3.3.1. Pre-Hospital Time

The time averaged was 7.0 ± 4.1 days (median/IQR: 7.0/4.0–9.0), observing cases that were hospitalized on the first day of the onset of symptoms up to cases that did so on day 21. The Kruskal-Wallis test showed that there was a statistically significant difference in the pre-hospital time according to the age group of adults (Dunn-Bonferroni test; *p* = 0.001) and seniors (Dunn-Bonferroni test; *p* = 0.002) who had a significantly longer time to arrive at the hospital from the onset of symptoms (Figure 1). Likewise, women arrived at the hospital in a significantly shorter time than men (U Mann-Withney test; *p* = 0.037). Other variables such as the region of origin and the healthcare provider did not significantly affect pre-hospital time (Table 4).

The multivariable analysis with a Poisson regression model found that the age group, comorbidity diagnosis and the region of origin significantly influenced pre-hospital time. In this manner, the seniors, adults, and young people had significantly greater pre-hospital time, while people with diagnosis of comorbidities and those from the Andean region had significantly less prehospital time (Table 5).

#### 3.3.2. Hospitalized Time

The time-averaged was 6.9 ± 6.0 days (median/IQR: 5.0/3.0–9.0); thus, some people died on the first day of admission to the hospital up to those who died 47 days after admission, after a prolonged stay that included admission to an intensive care unit and mechanical ventilation. The Kruskal-Wallis test showed a statistically significant difference in hospital time according to age group, with significantly shorter times in seniors than adults (Dunn-Bonferroni test; *p* < 0.001), indicating complications they died very quickly once they entered the hospital. A similar situation was observed in women that had complications and died in less time than men (Table 4).

There was a statistically significant difference in hospital time according to the region of origin (*p* < 0.001), observing in the post hoc analysis a significantly shorter hospital time in the other departments of the coast (Dunn-Bonferroni test; *p* < 0.001), the Andean region (Test Dunn-Bonferroni; *p* = 0.003) and Amazon (Test Dunn-Bonferroni; *p* = 0.004) compared to Lima, Peru’s capital (Table 4, Figure 1). The regions of the rest of the coast presented significantly less hospital time than other regions.

In the case of healthcare provider, it is observed that the cases treated in private clinics (Dunn-Bonferroni test; *p* < 0.001), hospitals of the National Police and Armed Forces (PNP/FF: AA) (Dunn-Bonferroni test; *p* = 0.010), and Insurance Social of Peru EsSalud (Dunn-Bonferroni test; *p* = 0.010), survived longer time than the people who were treated at Ministry of Health and regional governments hospitals (MINSA/GORE) (Figure 1). The cases treated in private clinics (Dunn-Bonferroni test; *p* < 0.041) and hospitals of PNP/FF: AA (Dunn-Bonferroni test; *p* = 0.038) survived longer time than the people who were treated at Essalud hospitals.

There was no statistically significant difference observed in hospital time in those with a diagnosis of comorbidity compared to those without (U Mann-Withney test; *p* = 0.624).
healthcare-10-02404-t004_Table 4Table 4Comparison of means and medians of pre-hospital and hospital time (days) with epidemiological characteristics of deaths from COVID-19 in Peru.CharacteristicPre-Hospital Time (Days)Hospital Time (Days)Mean ± Standard Deviation/Median (IQR)*p*-ValueMean ± Standard Deviation/Median (IQR)*p*-ValueAge Group



Children and adolescents ^Ψ^2.9 ± 1.7/3.0 (1.0–4.3)0.0028.4 ± 9.4/5.0 (3.0–10.0)<0.001Youth7.1 ± 6.4/7.0 (4.0–9.0)4.9 ± 3.5/4.0 (2.0–8.0)Adults7.1 ± 3.9/7.0 (4.0–9.0) *7.9 ± 6.9/6.0 (3.0–10.3)Seniors7.0 ± 4.2/7.0 (4.0–9.0) *6.4 ± 5.4/5.0 (3.0–8.0) ^μ^Gender



Male ^Ψ^7.1 ± 4.1/7.0 (4.0–9.0)0.0377.1 ± 5.6/7.0 (3.0–10.0)<0.001Female6.8 ± 4.1/6.0 (4.0–9.0) *6.3 ± 6.0/4.0 (2.0–8.0) *Region of origin



Lima ^Ψ^7.1 ± 4.3/7.0 (4.0–9.0)0.1567.9 ± 6.5/6.0 (3.0–10.0)<0.001Rest of the coast7.0 ± 4.2/7.0 (4.0–10.0)5.8 ± 5.4/4.0 (2.0–8.0) *Andean region6.6 ± 3.3/6.0 (4.0–8.0)6.5 ± 5.2/5.0 (3.0–9.0) * ΩAmazon7.4 ± 3.7/7.0 (5.0–9.0)6.3 ± 4.8/5.0 (3.0–9.0) * ΩHealthcare Provider



Public hospital of the MINSA/GORE ^Ψ^6.9 ± 4.1/6.0 (4.0–9.0)0.5966.4 ± 5.5/5.0 (2.0–9.0)<0.001EsSalud public hospital7.1 ± 4.1/7.0 (4.0–10.0)6.9 ± 6.0/5.0 (3.0–9.0) *Hospital of PNP/FF.AA6.8 ± 3.9/7.0 (2.8–9.0)8.3 ± 6.2/6.0 (3.0–10.0) * ^ф^Private clinics7.1 ± 4.6/6.5 (4.0–9.0)10.5 ± 9.0/8.5 (4.0–14.3) * ^ф^Comorbidities



Yes6.9 ± 4.1/7.0 (4.0–9.0)0.0537.1 ± 6.4/5.0 (3.0–9.0)0.624No ^Ψ^7.3 ± 4.2/7.0 (4.0–9.0)6.7 ± 5.5/5.0 (3.0–9.0)^Ψ^ Reference category, MINSA/GORE: Ministry of Health and regional governments, EsSalud: Insurance Social of Peru. * Statistically significant difference compared to the reference category. ^μ^ Statistically significant difference with the adult group. ^ф^ Significant difference with EsSalud. Ω Statistically significant difference with the Rest of the coast.

The multivariable analysis found that sex, comorbidity diagnosis, healthcare provider and the region of origin influenced hospital time. In this manner, masculine sex and the diagnosis of comorbidities increased hospital time significantly, while the medical care in MINSA/GORE, EsSalud, PNP/FF AA establishments, as well as Andean region and jungle origin, significantly reduced hospital time (Table 6). Age group and pre-hospital time did not significantly influence hospital time.
Figure 1(**A**) Comparison of the pre-hospital time of deaths from COVID-19 according to age group. (**B**) Comparison of hospitalized time (days) of cases of COVID-19 death by region of residence. (**C**) Comparison of hospitalized time (days) of cases of COVID-19 death by healthcare provider.
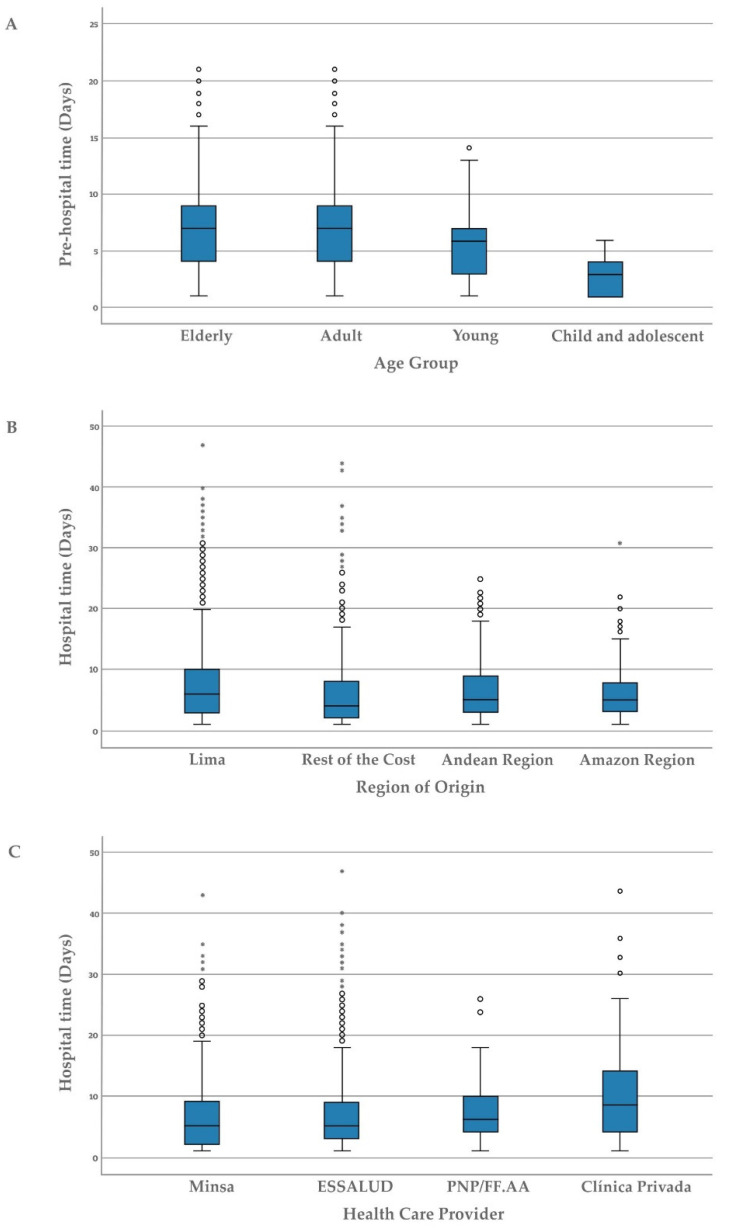

healthcare-10-02404-t005_Table 5Table 5Multivariable analysis of pre-hospital time of COVID-19 deaths during the initial pandemic response.ParametersB95% Wald Confidence IntervalContrast HypothesisInferiorSuperiorWald Chi-Squaredf*p* Value *Gender0.035−0.0240.0931.35910.244MaleFemale (Ref)Age Group





Seniors0.8660.5281.20425.2331<0.001Adults0.8730.5341.21225.4771<0.001Youth0.5010.0120.9914.03010.045Children and adolescents (Ref)





Comorbidities−0.058−0.110−0.0054.66210.031YesNo (Ref)Healthcare Provider





Public hospital of the MINSA/GORE−0.027−0.1870.1330.10810.743EsSalud public hospital0.020−0.1400.1800.05810.810Hospital of PNP/FF.AA−0.068−0.3060.1700.31410.575Private clinics (Ref)





Region of origin





Amazon0.036−0.0500.1220.66610.414Andean region−0.103−0.187−0.0195.77810.016Rest of the coast0.004−0.0540.0630.02210.883Lima (Ref)





* Poisson regression with robust variance. B: Beta coefficient. df: Degrees of freedom.
healthcare-10-02404-t006_Table 6Table 6Multivariable analysis of variables that influence on hospital time of COVID-19 deaths in Peru during the initial pandemic response.ParametersB95% Wald Confidence IntervalContrast HypothesisInferiorSuperiorWald Chi Squaredf*p* Value *Gender





Male0.1110.0180.2045.46010.019Female (Ref)





Age Group





Seniors0.022−0.3300.3730.01410.904Adults0.213−0.1410.5681.39110.238Youth−0.184−0.7290.3600.44110.507Children and adolescents (Ref)





Comorbidities





Yes0.0800.0010.1583.95710.047No (Ref)





Healthcare Provider





Public hospital of the MINSA/GORE−0.451−0.672−0.23016.0331<0.001EsSalud public hospital−0.341−0.562−0.1209.14610.002Hospital of PNP/FF.AA−0.274−0.5890.0412.91510.088Private clinics (Ref)





Region of origin





Amazon−0.150−0.288−0.0134.59410.032Andean region−0.105−0.2330.0242.55210.110Rest of the coast−0.242−0.334−0.15127.1401<0.001Lima (Ref)





Pre-hospital time





9 or more days−0.089−0.1960.0192.58910.1084 to 8 days−0.059−0.1600.0421.31810.2511 to 3 days (Ref)





* Poisson regression with robust variance. B: Beta coefficient. df: Degrees of freedom.


## 4. Discussion

This research shows that, almost three months after the first confirmed case of COVID-19 in Peru, deaths from COVID-19 were more frequent in males, seniors, residents of the region of Lima, and the rest of the Peruvian coast. A third of the deceased had comorbidities; likewise, a considerable fraction died at home, in accommodation, shelter, on a public road, or in transit to a hospital. Pre-hospital time was affected by age group, the diagnosis of comorbidities and the region of origin; while, hospital time was influenced by gender, the diagnosis of comorbidities, healthcare provider, and the region of origin.

Our findings found that the characteristics of COVID-19 deaths in Peru during the initial pandemic response are similar to those reported in Ecuador [20,21] (distribution by age and sex), a country that until 18 April 2020 constituted the Latin-American country hardest hit by the COVID-19 pandemic in terms of mortality, which is explained by its sociodemographic conditions and public health system capacity being similar to Peru during the initial pandemic response. They are also similar to those reported in Brazil [22,23] where, in addition, we observe a geographic pattern predominantly of deaths in the northern region of the country, where a lower health system response capacity existed. The mortality rate from COVID-19 in Peru as of 21 May 2020 (11.8 per 100,000 inhabitants) was higher than that reported by Ecuador [21] as of 18 April 2020 (2.7 per 100,000 people) and that reported by Brazil [23] as of May 16 2020 (7.4 per 100,000 inhabitants).

Deaths from COVID-19 were more frequent in seniors [24], similar to the study reported by Kang [25], who found an exponential increase in the death rate from COVID-19 as age increases, regardless of the geographical region. Changes in the immune system could also explain the greater vulnerability in seniors due to aging and innate and adaptive immunity (immunosenescence) [24,25,26]. Studies conducted in China show that the age-related pattern of death from COVID-19 differs from other respiratory viruses, where the severity pattern is often described as a U-shaped curve, with morbidity and mortality concentrated in groups of extreme age (young children and the seniors), affects adults, and is concentrated in seniors [25,27].

In the present series of cases, it is observed that seven out of every 10 deaths were men, with the greater severity of the disease [28,29]. This difference is not defined by age since women presented a statistically significant higher average age than men.

On the other hand, it was observed that a third of deaths from COVID-19 were associated with comorbidity, which could be explained by the greater severity and speed of the clinical picture observed in people with comorbidities that frequently lead to death [30,31]. The high proportion of deaths in older adults could also be explained by the higher frequency of comorbidities in this age group; thus, our research finds a significant increase in the proportion of comorbidities as age increases. The frequency of comorbidities reported in our study is almost double that reported in Ecuador [21] during the initial phase of the pandemic (31.4% versus 16.1%), which could explain, in part, the reasons why Peru was hit the hardest in terms of mortality.

Regarding the place of occurrence, 17.8% of deaths during the study period were out-of-hospital. The casualty occurred at the patients’ home, in accommodation or nursing home, shelter, on a public road, or in transit to a hospital. The proportion of out-of-hospital deaths is considerably higher than that observed on 31 March 2020, in which 0.2% of deaths confirmed by COVID-19 were out-of-hospital [32] and on 30 April 2020, where the proportion of out-of-hospital deaths accounted for 8.2% of total confirmed deaths [33]. This could be related to the collapse of health services, observing that the number of hospital beds, ICU beds, medical specialists, and health professionals available was largely insufficient [34,35,36]. The high proportion of out-of-hospital deaths that occurred in Peru, reflects the great impact of COVID-19 on health systems, as well as the inequity in the population’s access to health services, particularly in countries with fragmented health systems like Peru [37,38,39].

Other aspects that would explain the high proportion of home fatalities were self-medication and insufficient oxygen supply. At first, a significant fraction of the population preferred to self-medicate with hydroxychloroquine, azithromycin, ivermectin, and even chlorine dioxide, instead of going to a hospital, died in their homes due to the rapid evolution of the disease [40,41,42]. Later, the supply of oxygen proved to be largely insufficient in public health facilities in the interior of the country as they did not have oxygen plants, which led people to decide to manage their family members at home, acquiring oxygen from private providers combined with the exaggerated increase in the cost of this service, so many families could not afford it. It should be noted that a study carried out in a hospital in Lima found that oxygen saturation of less than 85% was the main predictor of death in people with COVID-19 pneumonia [43]. It is important to mention that during the initial response to the pandemic, Peru did not have vaccines against COVID-19 and vaccination began on 9 February 2021, during the second wave of the pandemic, with the prioritization of the Plan Nacional de Vacunación (national vaccination plan) of vulnerable populations such as healthcare workers, seniors, and people with comorbidities [44].

Pre-hospital care through rapid response teams, clinical follow-up teams, or medical care in first-level care facilities is essential for the identification, isolation, and treatment of cases, as well as for the early referral of cases to hospitals in the case of pneumonia, the rapid evolution of the clinical picture, or the presence of complications [45]. Our research shows that among the deaths due to COVID 19 there was a significant delay in going to a hospital among adults and seniors, while children and adolescents who were taken or referred to a hospital in a short time; similarly women sought care in less time, which could be related to an inadequate perception of the risk of severe disease, similar to the study by Chan reported in China [46].

On the other hand, hospital time was shorter in the rest of coast and Amazon regions where cases died quicker than in the region of Lima; this could be due to the lower response capacity of the hospitals in these departments compared to Lima. Age would not have had a significant influence on hospital time since stratified analysis found no significant differences in Lima, or the rest of the coast, Andes mountain range, and jungle. This has also been documented in countries such as Ecuador and Brazil, which experienced greater difficulties in cities located on the coast such as Guayaquil and in the Amazon region of Brazil, particularly in the State of Amazonas (which geographically borders with Peru, Colombia, and Venezuela), as a consequence of the health system collapse and funeral service crisis [21,22,47].

Private clinics and hospitals of the PNP/FF.AA showed longer survival time of cases, in contrast to the MINSA/GORE hospitals and social security establishments, which faced a high number of cases that far exceeded the resources provided. Countries such as China that were more efficient in improving their response capacity in the short term (an increase in the number of hospitals, the number of hospital beds, beds in intensive care units, and specialized health personnel) presented better results in the management of hospitalized cases, which resulted in the reduction in the number of deaths [48].

This research shows in part the deficiencies that led Peru to have the highest mortality from COVID-19 worldwide; however, we believe that broader analyzes should be carried out in order to design interventions adapted to the sociocultural context of the country, so that these results are not repeated in future epidemics or pandemics, by improving the early response to public health events of international importance.

Within the limitations, it must be considered that our study was carried out using secondary sources, so it is possible that there are quality problems and underreporting to some degree. For example, this could have occurred in variables such as the diagnosis of comorbidities, particularly in those who died outside a health facility, or did not have access to health services, or may have lost some degree or relevant clinical or epidemiological information. However, the fact of having considered several sources of information, as well as the verification and investigation of each death, would partially offset these limitations. Studies carried out in Italy [8] show that some of the out-of-hospital deaths may not have been recorded in the official COVID-19 registries at the peak of the epidemic, particularly deaths at home, or those that occurred in nursing homes.

Another limitation is that the molecular RT-PCR test is not considered a laboratory test in all cases, but a significant fraction of cases were diagnosed using rapid tests. Nevertheless, it should be considered that most of them presented a clinical picture compatible with coronavirus pneumonia, and a significant fraction was hospitalized, for this reason, making it difficult to classify the death being due to any other disease other than COVID-19.

Finally, for the present study, only deaths that were confirmed with laboratory test, that met the epidemiological criteria, and that presented a clinical picture compatible with COVID-19, were considered. It should be considered that during the peak of the first pandemic wave in Peru, a significant fraction of deaths were not confirmed with laboratory tests. These deaths were not included in our study because, in a significant proportion, it was not possible to assess prehospital and hospital times. Although the COVID-19 mortality rate obtained in this article is underestimated, it is possible to compare differences between regions during the initial pandemic response.

## 5. Conclusions

Deaths from COVID-19 occur mainly in males, seniors, in Lima residents, and other coastal departments, with considerable deaths at home, in shelters, on public roads, or in transit to a hospital. Pre-hospital time was affected by age group, the diagnosis of comorbidities, and the region of origin; while, hospital time is influenced by gender, the diagnosis of comorbidities, healthcare provider, and the region of origin.

## Figures and Tables

**Table 1 healthcare-10-02404-t001:** Epidemiological characteristics of the deaths confirmed by COVID-19 in Peru.

Characteristic	Frequency	%
Age Group		
Children and adolescents	18	0.5
Youth	26	0.6
Adults	1159	30.1
Seniors	2648	68.8
Gender		
Male	2772	72.0
Female	1079	28.0
Region of origin		
Lima	1645	42.7
Rest of the coast	1511	39.2
Andes mountain range	292	7.6
Jungle	403	10.5
Place of occurrence of the death		
Public hospital	3071	79.8
Private clinic	78	2.0
Home, accommodation, shelter, public road,	686	17.8
Transit to the hospital		
Penitentiary institution	16	0.4
Diagnosis of comorbidities		
Yes	1211	31.4
No	2640	68.6
Diagnosis with laboratory tests		
RT-PCR	1282	33.3
Rapid test IgM detection/IgG	2271	59.0
Both tests	298	7.7

**Table 2 healthcare-10-02404-t002:** The mortality rate from COVID-19 by region of Peru from March 28 to 21 May 2020.

Region of Origin	Mortality Rate from COVID-19 * (per 100,000 People)
Lima	14.0
Rest of the coast	18.1
Andes mountain range	3.1
Jungle	12.9
Peru	11.8

* Includes only deaths confirmed with laboratory tests.

**Table 3 healthcare-10-02404-t003:** Stratified analysis of the epidemiological characteristics of deaths confirmed by COVID-19 in Peru.

Variable	X¯ ± SD/Frequency (%)	*p*-Value
Gender		0.002
Male	64.9 ± 14.0	
Female	66.5 ± 15.0	
Region of origin		
Lima	65.6 ± 14.2	0.141
Rest of the coast	65.4 ± 14.3	
Andes mountain range	65.5 ± 13.6	
Jungle	63.8 ± 15.5	
Diagnosis of comorbidities by age categories		
Children and adolescents	3/17 (17.6%)	0.020
Youth	5/20 (25.0%)	
Adults	332/1026 (32.4%)	
Seniors	871/2358 (36.9%)	

## Data Availability

Data may be made available by contacting the corresponding author.

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
