# Peer review of "Epidemiological Characteristics of Deaths from COVID-19 in Peru during the Initial Pandemic Response"

_healthcare, 2022, doi:10.3390/healthcare10122404_

Round 1

Reviewer 1 Report

The ongoing COVID-19 pandemic has already caused 618 millions of infection and 6.55 million of SARS-CoV-2 infection- related deaths. It is extremely important to examine local, national and international COVID-19-caused death risk factors and implement the appropriate antiviral strategy. Ramos et al. in their manuscript described the COVID-19 deaths in the Peru population. The aim of this project is quite clear; however the manuscript needs some corrections (please see below). 

MAJOR COMMENTS:

1.     While the direct death numbers can be used for the identification of the death risk factors, authors should remember that from the epidemiological point of view the deaths number per 100 000 inhabitants is much more informative and descriptive parameter and should be used especially in the comparison of the geographic regions, i.e. Lima and jungle. I can imagine that the population size in the capital is much higher than in the jungle, so it is not surprising that there are more deaths accordingly. But it is incorrect to say that the COVID-19 deaths are more frequent in Lima than in the jungle! I would strongly recommend including the deaths number per 100 000 citizens in your manuscript and include the effect of the region on the COVID-19 deaths in the Discussion, including i.e., the healthcare system availability.

2.     Additionally, authors focused mostly on the deaths and did not show the number of the COVID-19 cases. I would strongly suggest including the daily new COVID-19 cases in Peru on your Figure 1. And additionally, since you compare the deaths between different regions, it would be beneficial to also compare the number of new SARS-CoV-2 infection geographically. And last but not the least, if you know the official number of COVID-19 cases and number of deaths, you can estimate the COVID-19 mortality and, as previously mentioned, compare it between different parts of Peru.

3.     Figure 1 – as far as I can understand, the line on the figure is an estimated trend of COVID-19 deaths. Why is the trend growing within last few days while the number of deaths dropped significantly? It is also mentioned by the authors, so the trend line should be corrected.

4.     Table 1 – the serological tests detecting the anti-SARS-CoV-2 IgG are used only in the retrospective analysis and cannot be used to detect the current infection!

5.     Discussion – I would strongly recommend to compare the COVID-19 cases and deaths in neighboring countries, as well as other comparable regions with similar population and healthcare systems.

6.     Discussion – the authors described the COVID-19 deaths based on the results of the RT-PCR and rapid cassette tests. It is commonly known that the rapid cassette tests are less sensitive and can give up to 50% of the false negative results. How this fact can biased your results?

7.     And finally, authors cite many Spanish documents which are not understandable for the non-Spanish speaking readers. Maybe it would be better to link the information with the official data presented on the WHO site.

MINOR COMMENTS

Page 2 Lines 54-59 – when the ranking was done? How can it be compared with other countries of this region? The citation is also missing.

Page 2 Lines 64-65 – it would be good to know how the Peruvian Ministry of Health tried to organize the response with the available resources. The Peruvian anti-COVID-19 strategy in the analyzed time period should be described in details.

Page 2 Line 73 – ‘Report of case series type’ – something is missing

Table 1 – the sum of the % in the diagnosis of comorbidities is not equal to 100%

Figure 2 – I would recommend increasing the font size on the figure. In addition, the figure lacks the legend for the subfugure C. 

Author Response

MAJOR COMMENTS:

  1. While the direct death numbers can be used for the identification of the death risk factors, authors should remember that from the epidemiological point of view the deaths number per 100 000 inhabitants is much more informative and descriptive parameter and should be used especially in the comparison of the geographic regions, i.e. Lima and jungle. I can imagine that the population size in the capital is much higher than in the jungle, so it is not surprising that there are more deaths accordingly. But it is incorrect to say that the COVID-19 deaths are more frequent in Lima than in the jungle! I would strongly recommend including the deaths number per 100 000 citizens in your manuscript and include the effect of the region on the COVID-19 deaths in the Discussion, including i.e., the healthcare system availability.

Response: Our study consists of case series that present the characteristics of COVID-19 deaths using frequency distributions, therefore, we consider that it is not incorrect to mention that one characteristic is more or less frequent in reported cases. Introducing incidences imply population comparisons, which was not the study’s objective, furthermore it would imply that the study change from a case series to a cohort study or ecological study, which also was not the study’s objective.

  1. Additionally, authors focused mostly on the deaths and did not show the number of the COVID-19 cases. I would strongly suggest including the daily new COVID-19 cases in Peru on your Figure 1. And additionally, since you compare the deaths between different regions, it would be beneficial to also compare the number of new SARS-CoV-2 infection geographically. And last but not the least, if you know the official number of COVID-19 cases and number of deaths, you can estimate the COVID-19 mortality and, as previously mentioned, compare it between different parts of Peru.

Response: Our study constitutes a case series. Our objective is not to compare populations or describe the behavior of SARS-CoV-2 infections, rather, to describe COVID-19 death characteristics at the beginning of the pandemic in Peru. The case series studies only include a group of people or patients with some characteristics that one wishes to describe and introducing the number of cases would distort the design used to characterize COVID-19 deaths.

  1. Figure 1 – as far as I can understand, the line on the figure is an estimated trend of COVID-19 deaths. Why is the trend growing within last few days while the number of deaths dropped significantly? It is also mentioned by the authors, so the trend line should be corrected.

Response: The number of deaths dropped because a fraction of them were later confirmed by delays in the process of laboratory tests, this is why we observe a decrease in the number of deaths in the last two weeks. The authors have made an effort to recover the deaths that are confirmed extemporaneously, recovering 607additional deaths, unfortunately, we did not have this data when we completed the first version of the manuscript. However, we have removed figure 1 to prioritize tables 4 and 5 of the multivariate analysis from pre-hospitalization and hospitalization time requested by one of the reviewers.  

  1. Table 1 – the serological tests detecting the anti-SARS-CoV-2 IgG are used only in the retrospective analysis and cannot be used to detect the current infection!

Response: At the beginning of the pandemic, the Peruvian State acquired IgM/IgG serologic tests, thanks to the IgM it was possible to discriminate recent infections from past infections, which was additionally correlated with the respiratory clinical evidence of patients.

  1. Discussion – I would strongly recommend to compare the COVID-19 cases and deaths in neighboring countries, as well as other comparable regions with similar population and healthcare systems.

Response: In the discussion, we made the comparison with Ecuador and Brazil from available existing publications.

  1. Discussion – the authors described the COVID-19 deaths based on the results of the RT-PCR and rapid cassette tests. It is commonly known that the rapid cassette tests are less sensitive and can give up to 50% of the false negative results. How this fact can biased your results?

Response: It is correct that rapid tests have less sensitivity, and the possibility of false negatives exist, however, during the first months (in the first wave) when the availability of molecular tests notably decreased in Peru, rapid IgM/IgG tests were the only ones available, and they were acquired and distributed by the Peruvian State. If the matter were case testing it would have been a great problem, but in this case, deaths that had a COVID-19 pneumonia clinical presentation (clinical criteria) are described and an important percentage of them also had contact with other cases (epidemiological criteria). Since the number of deaths was manageable, these were registered and verified by Peruvian epidemiologists. While it is certain that rapid test use is not the best option, we believe that this did not significantly affect the results. These limitations have been addressed in the discussion of results.

  1. And finally, authors cite many Spanish documents which are not understandable for the non-Spanish speaking readers. Maybe it would be better to link the information with the official data presented on the WHO site.

Response: We have made every effort to consider the greatest amount of literature in English. The sources in Spanish correspond to data in Peru that do not have a counterpart in the WHO website.

MINOR COMMENTS

Page 2 Lines 54-59 – when the ranking was done? How can it be compared with other countries of this region? The citation is also missing.

Response: We have provided ranking clarifications and have added the bibliographic reference.

Page 2 Lines 64-65 – it would be good to know how the Peruvian Ministry of Health tried to organize the response with the available resources. The Peruvian anti-COVID-19 strategy in the analyzed time period should be described in details.

Response: We provided details on the response of the Peruvian State to interrupt the SARS-CoV-2 transmission chain.

Page 2 Line 73 – ‘Report of case series type’ – something is missing

Response: The sentence has been corrected.

Table 1 – the sum of the % in the diagnosis of comorbidities is not equal to 100%

Response: After reviewing and adding the deaths registered in an extemporaneous manner, the percentages were recalculated, adding up to 100% this time.

Figure 2 – I would recommend increasing the font size on the figure. In addition, the figure lacks the legend for the subfugure C. 

Response: The font size was increased, and item C was added to the legend of Figure 1.

Reviewer 2 Report

This study summarizes the epidemiological characteristics of death from COVID-19 in Peru from March to May 2020. The country was affected by the SARS-CoV-2 and had the highest mortality rate from COVID-19 as of the end of August 2020. It is crucial to learn from the situation that the country had to prepare for the future pandemic; thus this kind of study should be conducted. However, there are several flaws in the method of this study to investigate the risk factors for the pre-hospitalized and post-hospitalized periods.

First, the authors analyzed good demographic information on the patients who died; however, only univariable analyses were conducted without any control of confounders such as age, sex, and reasons (although each of them was analyzed separately in each univariable analysis). Multivariable regression analyses should be applied to them, or at least stratified analyses for each potential confounder should be performed. Age-stratified analysis was done for the comorbidity conditions in the study; this kind of approach can be applied to pre- and post-hospitalized periods although multivariable analysis is preferable for the analysis with multiple confounders.

Second, the post-hospitalized period must have been affected by the pre-hospitalized period because if the time until hospitalisation since symptom onset was too long due to a shortage of beds, the time to death after hospitalization would be short. Therefore, when evaluating the factors on hospital time, the pre-hospital time should be controlled (e.g. using multivariable analysis).

Third, the data of death used in this study must have been right-censored, i.e., cases who are diagnosed as SARS-CoV-2 positive close to May 2021 (cut-off date) but died after the cut-off date were not included due to the lack of observation time. Therefore, the hospital time measured using individuals who died by May 21, 2020, might be underestimated. To avoid this bias, you can remove cases without full observational time for death, by removing cases who are diagnosed as positive a certain period earlier than May 21.

Author Response

First, the authors analyzed good demographic information on the patients who died; however, only univariable analyses were conducted without any control of confounders such as age, sex, and reasons (although each of them was analyzed separately in each univariable analysis). Multivariable regression analyses should be applied to them, or at least stratified analyses for each potential confounder should be performed. Age-stratified analysis was done for the comorbidity conditions in the study; this kind of approach can be applied to pre- and post-hospitalized periods although multivariable analysis is preferable for the analysis with multiple confounders.

Response: We appreciate Reviewer 2’s recommendation. We added the multivariate analysis of pre-hospitalization time and of hospitalization time for which we have used a Poisson regression model with robust variance.

Second, the post-hospitalized period must have been affected by the pre-hospitalized period because if the time until hospitalisation since symptom onset was too long due to a shortage of beds, the time to death after hospitalization would be short. Therefore, when evaluating the factors on hospital time, the pre-hospital time should be controlled (e.g. using multivariable analysis).

Response: We included pre-hospitalization time within the hospitalization time analysis, although we did not find a statistically significant relationship.

Third, the data of death used in this study must have been right-censored, i.e., cases who are diagnosed as SARS-CoV-2 positive close to May 2021 (cut-off date) but died after the cut-off date were not included due to the lack of observation time. Therefore, the hospital time measured using individuals who died by May 21, 2020, might be underestimated. To avoid this bias, you can remove cases without full observational time for death, by removing cases who are diagnosed as positive a certain period earlier than May 21.

Response: The authors have made an effort to recover deaths that were confirmed in an extemporaneous manner, recovering 607 additional deaths, unfortunately we did not have this data at the time the study was carried out.

Reviewer 3 Report

As we globally try to understand the epidemiology of COVID-19, this case study from Peru provides information about what was happening in that country at an early point in the epidemic. The data are presented nicely, but I would have liked to see a Table on cause of death by region. Yes, the virus contributed to death, but what was recorded on the death certificates? Respiratory failure? Cardiac arrest? What? I dont find the findings at all surprising and in line with that seen in other countries.

It also would be nice to know when the vaccine was made available in Peru. Is the reader to presume that no vaccinations were possible during this study period? A bit more on this would be helpful.

Why did the authors select such a short period of time (March 28-May 1 2020)? What is the rationale for this? 

Do the authors know if those hospitalized followed a treatment protocol? Did treatment options differ among the geographic areas? I would presume that Lima would have more advanced options compared to the Andean or jungle  regions, for example. Did this make a difference in mortality? More deaths were recorded in Lima primarily because the population is greatest there. 

Author Response

As we globally try to understand the epidemiology of COVID-19, this case study from Peru provides information about what was happening in that country at an early point in the epidemic. The data are presented nicely, but I would have liked to see a Table on cause of death by region. Yes, the virus contributed to death, but what was recorded on the death certificates? Respiratory failure? Cardiac arrest? What? I dont find the findings at all surprising and in line with that seen in other countries.

Response:  The authors appreciate the recommendation. Since Peru has 25 regions in addition to Metropolitan Lima we consider that Table 1 would be more extensive and that it would be difficult to interpret for those that do not know the regional organization of Peru. That is why we opted for the distribution according to natural regions: Lima (coast), remainder of the coast, mountain and jungle. The deaths included correspond to the deaths from pneumonia which is specified in the methodology of this article. At that moment it was difficult to determine deaths from causes other than COVID-19 given that there was not sufficient evidence (such as heart failure or ischemic diseases). The death certifications in the vital statistics system does not consider cardiac arrest as a basic cause or as a contributor to death.

It also would be nice to know when the vaccine was made available in Peru. Is the reader to presume that no vaccinations were possible during this study period? A bit more on this would be helpful.

Response: It has been added to the discussion section that, during the initial pandemic response, Peru did not have a vaccine against COVID-19 and vaccination began February 9, 2021.

Why did the authors select such a short period of time (March 28-May 1 2020)? What is the rationale for this?

Response: We selected this period because we wanted to focus our response on the beginning of the pandemic, which had this been better from the beginning, the country would have had better results which would have been reflected by a lower  number of COVID-19 deaths.

Do the authors know if those hospitalized followed a treatment protocol? Did treatment options differ among the geographic areas? I would presume that Lima would have more advanced options compared to the Andean or jungle  regions, for example. Did this make a difference in mortality? More deaths were recorded in Lima primarily because the population is greatest there.

Response: We have added to the discussion that the hospitalization time was less in the country’s inland region due to its lower response capacity  in human resources, hospital beds, ICU beds, supply of medical oxygen and mechanical ventilator.  

Round 2

Reviewer 2 Report

1) Third, the data of death used in this study must have been right-censored, i.e., cases who are diagnosed as SARS-CoV-2 positive close to May 2021 (cut-off date) but died after the cut-off date were not included due to the lack of observation time. Therefore, the hospital time measured using individuals who died by May 21, 2020, might be underestimated. To avoid this bias, you can remove cases without full observational time for death, by removing cases who are diagnosed as positive a certain period earlier than May 21.

Response: The authors have made an effort to recover deaths that were confirmed in an extemporaneous manner, recovering 607 additional deaths, unfortunately we did not have this data at the time the study was carried out.

>> Regarding the comment I left last time, censoring data is a crucial issue for this analysis. Please consider this issue seriously. The treatment of this is very easy to be done (i.e., simply remove data with the potential of censoring).

Were the 607 cases who died after the cut-off date diagnosed as SARS-CoV-2 positive before the cut-off date (May 21)? Those cases must have a longer hospitalization time, thus not including them leads to the underestimation of the hospital-death time. To avoid this bias, you can simply remove dead cases who were diagnosed a certain period earlier than May 21 from your analysis. This period can be, say, 97.5 percentile of the hospitalized-death time distribution of your data. For example, if it is 14 days, by removing cases who died between May 7 (May 21 minus 14 days) and May 21 from the analysis, you could avoid the censoring bias. The same issue should be dealt to pre-hospital time analysis, too.

2) Lines 119-121:

>> The reason why the multivariable analysis with Poisson regression with robust variance was used is not clear. To my understanding, regression with Gamma error analysis is suitable for time data (pre-hospitalization and hospitalization time). Please refer to the document (Hogg R V., Craig AT. Some special distributions. Introd. to Math. Stat. 4th ed., New York: Macmillan Publishing Co., Inc.; 1978, p. 90–121.). Please add a reference if Poisson regression with robust variance is suitable for the analyses in the current study.

3) Section 3.2 & Table 2:

Sorry that I did not point out this last time. “Gender” and “region of origin” are not age-stratified but they are the age that was stratified by gender and region (while “diagnosis of comorbidities” is an age-stratified analysis actually). Please consider deleting “age-“ from the section title and “age-stratified” from the header of Table 2 (i.e. they would be “3.2. Stratified analysis”, and “Variable”)

4) Figure 2:

Since Figure 1 was removed. Figure 2 now should be Figure 1.

5) the term multivariate analysis:

I think the term “multivariable” is more suitable for the analysis than “multivariate”. Please refer to the document (doi: 10.2105/AJPH.2012.300897).

(Just comment, no need to respond) We included pre-hospitalization time within the hospitalization time analysis, although we did not find a statistically significant relationship.

>> By including the pre-hospitalization time, the direct effect of factors analysed on the hospitalization time (without the effect of pre-hospitalization time) can be estimated no matter whether that pre-hospitalization is statistically significant or not.

Author Response

Responses to Reviewer 2

Reviewer 2: Regarding the comment I left last time, censoring data is a crucial issue for this analysis. Please consider this issue seriously. The treatment of this is very easy to be done (i.e., simply remove data with the potential of censoring).

Were the 607 cases who died after the cut-off date diagnosed as SARS-CoV-2 positive before the cut-off date (May 21)? Those cases must have a longer hospitalization time, thus not including them leads to the underestimation of the hospital-death time. To avoid this bias, you can simply remove dead cases who were diagnosed a certain period earlier than May 21 from your analysis. This period can be, say, 97.5 percentile of the hospitalized-death time distribution of your data. For example, if it is 14 days, by removing cases who died between May 7 (May 21 minus 14 days) and May 21 from the analysis, you could avoid the censoring bias. The same issue should be dealt to pre-hospital time analysis, too.

Response: We regret not having provided enough details. The issue with the 607 cases we added was that, while they were diagnosed and died prior to May 21, 2020, the confirmatory test results were released with a date later than May 21st, for this reason, they were not initially considered as COVID-19 deaths in our study.   Afterwards, we were able to access their data and recover them, verify their lab tests, dates of hospitalization and death, which is why there was no loss of data, there was no need for censorship, all their information was recovered. For this reason, we did not underestimate the hospital-death time.   

Reviewer 2: The reason why the multivariable analysis with Poisson regression with robust variance was used is not clear. To my understanding, regression with Gamma error analysis is suitable for time data (pre-hospitalization and hospitalization time). Please refer to the document (Hogg R V., Craig AT. Some special distributions. Introd. to Math. Stat. 4th ed., New York: Macmillan Publishing Co., Inc.; 1978, p. 90–121.). Please add a reference if Poisson regression with robust variance is suitable for the analyses in the current study.

Response: We added the following references related to the count-data analysis to evaluate the use of healthcare services, including time, that refer to the use of Poisson regression. The robust focus was carried out considering possible deviations from normality and some degree of overdispersion.

Salinas-Rodríguez A, Manrique-Espinoza B, Sosa-Rubí SG. Statistical analysis for count data: Use of healthcare services applications. Salud Publica Mex 2009;51:397-406.

Cotter Salvado J. The Determinants of Health Care Utilization in Portugal: An Approach with Count Data Models. Swiss J. Econ. Stat. 2008;144(3):437-458.

The reference consulted (Hogg R V., Craig AT. Some special distributions. Introd. to Math. Stat. 4th ed., New York: Macmillan Publishing Co., Inc.; 1978, p. 90–121.) also includes Poisson regression within the special distributions.

Reviewer 2: Sorry that I did not point out this last time. “Gender” and “region of origin” are not age-stratified but they are the age that was stratified by gender and region (while “diagnosis of comorbidities” is an age-stratified analysis actually). Please consider deleting “age-“ from the section title and “age-stratified” from the header of Table 2 (i.e. they would be “3.2. Stratified analysis”, and “Variable”)

Response: We made the corrections according to the reviewer’s recommendation.

Reviewer 2: Since Figure 1 was removed. Figure 2 now should be Figure 1.

Response: We made the corrections according to the reviewer’s recommendation.

Reviewer 2: I think the term “multivariable” is more suitable for the analysis than “multivariate”. Please refer to the document (doi: 10.2105/AJPH.2012.300897).

Response: We made the corrections according to the reviewer’s recommendation.

Reviewer 2: (Just comment, no need to respond) We included pre-hospitalization time within the hospitalization time analysis, although we did not find a statistically significant relationship.

By including the pre-hospitalization time, the direct effect of factors analysed on the hospitalization time (without the effect of pre-hospitalization time) can be estimated no matter whether that pre-hospitalization is statistically significant or not.

Response: The authors agree with this idea, the multivariable analysis allows us to obtain the independent effect of each factor without the effect of other variables.